# Optimization Method of Human Posture Recognition Based on Kinect V2 Sensor

**DOI:** 10.3390/biomimetics10040254

**Published:** 2025-04-21

**Authors:** Hang Li, Hao Li, Ying Qin, Yiming Liu

**Affiliations:** 1School of Information Engineering, Xi’an University, Xi’an 710065, China; lhkelvin@163.com; 2Key Laboratory of Spectral Imaging Technology, Xi’an Institute of Optics and Precision Mechanics, Xi’an 710119, China; liuyiming21@mails.ucas.ac.cn; 3Xi’an Longviews Electronic Engineering Co., Ltd., Xi’an 710048, China; qy1290953720@163.com

**Keywords:** human–computer interaction, human action recognition, bone point optimization

## Abstract

Human action recognition aims to understand human behavior and is crucial in enhancing the intelligence and naturalness of human–computer interaction and bionic robots. This paper proposes a method to improve the complexity and real-time performance of action recognition by combining the Kinect sensor with the OpenPose algorithm, the Levenberg–Marquardt (LM) algorithm, and the Dynamic Time Warping (DTW) algorithm. First, the Kinect V2 depth sensor is used to capture color images, depth images, and 3D skeletal point information from the human body. Next, the color image is processed using OpenPose to extract 2D skeletal point information, which is then mapped to the depth image to obtain 3D skeletal point information. Subsequently, the LM algorithm is employed to fuse the 3D skeletal point sequences with the sequences obtained from Kinect, generating stable 3D skeletal point sequences. Finally, the DTW algorithm is utilized to recognize complex movements. Experimental results across various scenes and actions demonstrate that the method is stable and accurate, achieving an average recognition rate of 95.94%. The method effectively addresses issues, such as jitter and self-occlusion, when Kinect collects skeletal points. The robustness and accuracy of the method make it highly suitable for application in robot interaction systems.

## 1. Introduction

The rapid advancement of artificial intelligence has spurred the development of numerous mission-critical applications, including industrial automation, bionic robotics, intelligent transportation, and virtual/augmented reality [1,2,3]. These applications require more natural and seamless human–computer interactions, making pose estimation in computer vision increasingly vital [4]. Natural human–computer interaction is a foundational technology for such applications, relying on intelligent sensors for data acquisition, transmission, and processing [5,6]. To meet the needs of these fields and better guide the behavioral actions of bionic robots, it is necessary to accurately recognize human actions in real time and achieve intuitive and effective human–robot interaction.

Human action recognition is a critical technology for achieving natural human–computer interaction [7] and has long been a prominent research focus in computer vision and related fields [8]. As one of the key enabling technologies for artificial intelligence, it finds applications in diverse domains, such as intelligent defense and control, robotics, medical information technology, and action analysis, among others [9]. The first step in achieving human–machine recognition is to collect human body data through intelligent sensors. This data are then transmitted to other intelligent devices for processing, enabling data computation and classification. Currently, the primary method for action recognition data acquisition relies on computer vision, which can be further divided into two main approaches [10]. One approach involves using the traditional color camera for information acquisition and processing recognition. The other utilizes three-dimensional depth intelligent sensors, such as Microsoft’s Kinect intelligent sensor (the manufacturer is Microsoft, from Redmond, United States), for data acquisition. These kinds of intelligent sensors are not affected by environmental factors, such as lighting, and allow users to easily acquire color images, depth images, and other information related to human skeletal points. In the literature [11], a simple yet effective gait recognition algorithm based on human silhouettes was proposed. This method utilizes color images for segmentation, feature extraction, and other operations, and achieves recognition by comparing the results with a standard database. However, such approaches relying on traditional color cameras are susceptible to environmental factors, such as illumination, and the quality of the captured images may also exhibit certain limitations [12]. In literature [13], a Kinect sensor was employed for human data acquisition. This approach segments the human body into parts for joint localization and performs action recognition through joint information, demonstrating strong robustness. Compared to methods based on traditional RGB sensors, recognition techniques utilizing depth sensors are more accurate and less constrained by environmental factors. Therefore, we have conducted further research on these depth-based methods.

Depth-based sensors, such as Kinect and 3D skeletal point recognition methods, are unaffected by complex environmental factors, such as illumination. The Kinect sensor obtains skeletal points through operations, such as body segmentation, edge detection [14], and noise processing [15], and uses a random decision forest [16] as a feature classifier for training, allowing the obtained skeletal points to have higher recognition accuracy and greater robustness. The literature [17] proposed a posture recognition method based on distance characteristics. The Kinect was used for depth images and human data acquisition, which were then transformed into distance eigenvectors and normalized. Action recognition was achieved through a static posture recognition method based on the shortest distance and a dynamic posture recognition method employing dynamic time warping with minimum cumulative distance. The recognition rates for static poses, cross-subject dynamic poses, and non-cross-subject dynamic poses reached 95.9%, 89.8%, and 93.6%, respectively. However, since the method relied on the shortest distance for classification, each posture could only be assigned to a single category, leading to potential misclassification errors. To address this limitation, the K-Nearest Neighbor (KNN) algorithm can be employed to enhance recognition accuracy [18]. The literature [19] proposed a virtual image-based interaction system for the Kinect V2 sensor’s (the manufacturer is Microsoft, from Redmond, United States) inability to track the ankle joint. The system used a Kinect V2 sensor to capture body information, combined with a Shimmer Inertial Measurement Unit (IMU) sensor (the manufacturer is Shimmer, from Dublin, Ireland.) placed on the patient’s foot, effectively completing the body and ankle motion tracking. This system can assist in rehabilitating patients who have undergone total knee replacement surgery. However, the system has limited feedback capabilities and needs to be strengthened. The literature [20] used Kinect sensors to obtain body posture data from physical education teachers in a simulated classroom environment. By integrating classical machine learning algorithms, the study achieved intelligent recognition of classroom teaching behaviors exhibited by physical education teachers. Kinect 1.0 was used to test 10 physical education teachers in a simulated classroom. Through time sampling, body posture characteristics corresponding to different teaching behaviors of physical education teachers were extracted. The Light Gradient Boosting Machine (LightGBM) recognition model, combined with the Kinect sensor, achieved an accuracy of 0.998, significantly outperforming other algorithms. However, the study was limited to recognizing the teaching behaviors of a single individual in a controlled environment and did not support the recognition of behaviors involving multiple individuals.

However, relying solely on depth sensors for human recognition has certain limitations, such as skeletal point jitter or self-occlusion issues. To address these challenges, integrating the OpenPose algorithm for data fusion can be a viable solution. OpenPose is an open-source multi-person pose estimation algorithm using the Convolutional Neural Network (CNN) [21] and the Caffe framework proposed by Cao et al. [22] from Carnegie Mellon University. The literature [23] presents a TSSCI thesis paper on 2D OpenPose skeleton point detection, which relies exclusively on RGB images. The integration with Kinect provides complementary 3D positional data for skeletal joints, enhancing the overall framework. The potential benefits of combining the other 3D modalities with the 2D OpenPose output for TSSCI are significant. Improves the accuracy and robustness of TSSCI-generated motion capture. The literature [24] used a Kinect sensor to acquire 3D skeletal point data, and the OpenPose and LM algorithms are combined for data processing to obtain stable sequences. This method effectively improves recognition stability. However, its recognition is slow and limited to verifying simple actions, such as raising a hand, and it cannot recognize other or more complex actions. The literature [25] proposed a method for teaching trajectory planning based on human pose estimation. The trajectory video is captured using the Kinect sensor, and the OpenPose algorithm is employed to process the trajectory information. The results demonstrate that this method is more stable and accurate compared to using the Kinect sensor or OpenPose alone. However, the recognition success rate is only 73.4%, which remains relatively low. The literature [26] integrates an RGB-D camera and the OpenPose algorithm to identify 3D body landmark locations. Racking errors were examined and compared using two RGB-D cameras with different data acquisition techniques (stereo and time of flight (ToF)). The average tracking errors were 7.96 ± 3.59 and 9.81 ± 5.57 cm (stereo) for landmark positions and 6.38 ± 2.88 and 8.18 ± 5.56 cm (ToF) for standing and sitting postures, respectively. The study has some limitations, requiring specific correction for each fixation bias, and the results can be affected by the body size of the experimenter.

A multi-algorithm co-optimization approach is proposed to address the issues of low accuracy and self-occlusion of Kinect sensors in motion recognition. The Kinect V2 sensor is used to acquire the color image, depth image, and 3D skeleton point data of the human body, and combined with the 2D skeleton points extracted by the OpenPose algorithm for coordinate mapping and transformation, to achieve the spatial alignment of 3D skeleton data from multiple sources. To address the limitations of traditional neural network methods, the LM algorithm is used for nonlinear optimization of fused data. By adding appropriate constraints and constraint coefficients to the LM algorithm, faster and more stable results can be obtained. The stability of the skeleton point sequence can be effectively improved. Concurrently, the DTW algorithm is incorporated to construct the action spatio-temporal feature matching model, thereby enhancing the accuracy and real-time performance of complex action recognition [27]. The experimental findings demonstrate that the multi-algorithm co-optimization method effectively addresses the Kinect issues and exhibits a superior recognition rate and accuracy for complex actions. With enhanced accuracy and expedited recognition, this approach can be more efficaciously implemented in sophisticated scenarios, such as smart healthcare and robotics [28].

In order to further validate the advantages of the proposed multi-algorithm co-optimization method, a comparative analysis was conducted. A systematic comparison of the method with the classical scheme is conducted from multiple perspectives, including algorithm architecture, feature extraction, and optimization mechanism.

The validation results are presented in Table 1. The conventional approach is characterized by its reliance on extensive, labeled data sets for training, which can be costly and time consuming. Furthermore, the applicability of this method is limited. Compared with the conventional approach, the proposed method has been shown to enhance the robustness and recognition rate of action recognition through multi-source complementation and multi-algorithm co-optimization. Also, the efficiency of individual algorithms can be optimized to improve the overall efficiency. This approach offers a more stable realization method for bionic robots, rehabilitation, and medical scenarios.

## 2. Skeletal Points Processing Method

### 2.1. Skeletal Points Information

The human body exhibits a complex morphological structure, with significant variations in physique among individuals. To better analyze human motion dynamics, it is essential to simplify the human body based on Newtonian mechanics. This simplified model eliminates unnecessary tissues and organs, retaining only the skeleton and joints, and treats the skeleton as a rigid body [29]. The Kinect V2 sensor can capture 25 3D skeletal points of the human body, as illustrated in Figure 1a, while the OpenPose algorithm extracts 18 2D skeletal points, as shown in Figure 1b. These two sets of skeletal points exhibit notable differences. Since two methods were needed in this paper, the skeletal point sequences used for this purpose were also obtained by comparing and filtering the two skeletal point sequences, as shown in Figure 1c.

Since this study focuses on recognizing human movements and does not address hand and foot position detection, 15 skeletal points were purposely selected for analysis. The skeletal points represented by the numbers in the skeletal point sequence are as follows: 0-spine base, 1-spine shoulder, 2-head, 3-left shoulder, 4-left elbow, 5-left hand, 6-right shoulder, 7-right elbow, 8-right hand, 9-left hip, 10-left knee, 11-left foot, 12-right hip, 13-right knee, and 14-right foot.

### 2.2. Processing of Skeletal Points

Data fusion requires the acquisition of two types of 3D skeletal point sequences. First, a Kinect sensor was used to obtain a color image, a depth image, and 3D skeletal point coordinates P_k_ of the human body. The color image was then processed with the OpenPose algorithm to obtain the 2D skeletal point coordinates P_op_. To obtain another 3D skeletal point coordinates P_o_, it is necessary to map P_op_ with the depth image [30], and the mapping steps are as follows:

(1) Perform the internal reference calibration for the built-in color camera and depth camera of the Kinect sensor, and obtain the intrinsic matrices of the two cameras K_RGB_ and K_D_, as shown below:(1)$$K_{RGB}=\left[\begin{array}{ccc} f_{x\_RGB} & 0 & c_{x\_RGB}\\ 0 & f_{y\_RGB} & c_{y\_RGB}\\ 0 & 0 & 1 \end{array}\right]$$(2)$$K_{D}=\left[\begin{array}{ccc} f_{x\_D} & 0 & c_{x\_D}\\ 0 & f_{y\_D} & c_{y\_D}\\ 0 & 0 & 1 \end{array}\right]$$
where (f_x_RGB_, f_y_RGB_) and (c_x_RGB_, c_y_RGB_) are the focal length and center point coordinates of the color camera, respectively, and (f_x_D_, f_y_D_) and (c_x_D_, c_y_D_) are the focal length and center point coordinates of the depth camera, respectively.

(2) Calibrate the transformation relationship between the two cameras to obtain the rotation matrix of the depth camera coordinate system to the color camera coordinate system as R_D-RGB_ and the translation vector as t_D-RGB_.

(3) According to the intrinsic matrix of the depth camera, the 2D coordinates of the depth image can be mapped to the depth camera coordinate system to obtain the 3D coordinates. Taking a point in the depth image noted as (x_D_, y_D_) and the depth value as depth (x_D_, y_D_), then the 3D coordinates (X_D_, Y_D_, Z_D_) of the point in the depth camera coordinate system are as follows:(3)$$\left\{\begin{array}{c} X_{D}=(x_{D}-c_{x\_D})\times \frac{depth(x_{D},y_{D})}{f_{x\_D}}\\ Y_{D}=(y_{D}-c_{y\_D})\times \frac{depth(x_{D},y_{D})}{f_{y\_D}}\\ Z_{D}=depth\left(x_{D},y_{D}\right)\;\;\;\;\;\;\;\;\;\;\;\;\;\;\;\;\;\;\;\;\; \end{array}\right.$$

(4) Combining Equation (3), convert the 3D coordinates (X_D_, Y_D_, Z_D_) under the depth camera coordinate system to the 3D coordinates (X_RGB_, Y_RGB_, Z_RGB_) under the color camera coordinates as follows:(4)$$\left[\begin{array}{c} X_{RGB}\\ Y_{RGB}\\ Z_{RGB} \end{array}\right]=R_{D-RGB}\left[\begin{array}{c} X_{D}\\ Y_{D}\\ Z_{D} \end{array}\right]+t_{D-RGB}$$

(5) Furthermore, the 3D coordinates (X_RGB_, Y_RGB_, Z_RGB_) under the color camera coordinates in the color image coordinate system (x_RGB_, y_RGB_) are as follows:(5)$$\frac{1}{Z_{RGB}}\left[\begin{array}{c} x_{RGB}\\ y_{RGB}\\ 1 \end{array}\right]=K_{RGB}\left[\begin{array}{c} X_{RGB}\\ Y_{RGB}\\ Z_{RGB} \end{array}\right]$$

The color values corresponding to the coordinates (x_RGB_, y_RGB_) in the color image are taken as the color values of the 3D coordinates (X_RGB_, Y_RGB_, Z_RGB_).

(6) Repeat steps (3) to (5) for each point in the depth image.

In this way, the mapping from 2D to 3D is completed, and the second 3D skeletal point coordinates P_o_ can be obtained.

Then the above 3D skeletal point coordinates P_o_ are aligned with the 3D skeletal point coordinates P_k_ obtained from Kinect by translation. Firstly, the base point of the spine of the two models is used as the root node d. Secondly, the x and y direction coordinates at the two root nodes are differenced to obtain the difference value. Finally, all coordinate points of P_o_ are solved separately according to the difference value to obtain the translation data, i.e., the aligned data.

After the above processing, the two required 3D skeletal points are obtained and used as input data for fitting optimization to complete data fusion.

## 3. Skeletal Points Fitting Method

### 3.1. Fitting Principle

Suppose that there are two points in the three-dimensional coordinate system, the coordinates of one point are known, the coordinates of the other point can be obtained by the direction vector and distance of the two points, and the required optimization principles are the same. Take a left upper limb skeletal point as an example; suppose that the coordinates of the left shoulder, left elbow, and left hand are P_ls_, P_le,_ and P_lh_(x_h_, y_h_, z_h_), then the vector from the left elbow to the left shoulder is $\underset{T_{lels}}{\to }$ and the vector from the left elbow to the left hand is $\underset{T_{lelh}}{\to }$. To obtain the coordinates P_le_ and P_lh_ of the optimized left elbow and left hand, it is necessary to know the distance L_lsle_ and L_lelh_ from the optimized left shoulder to the left elbow and from the left elbow to the left hand, the details of the left upper limb skeletal points and their coordinate systems are shown in Figure 2. The algorithm used is to obtain the most suitable distance information by fitting the energy value E of Euler angles between skeletal points and then obtaining the new 3D skeletal point coordinates. To obtain better results, the Euler angles θ of multiple points are fitted simultaneously, which means that the optimal solutions of L_lsle_ and L_lelh_ are evaluated simultaneously to obtain the optimal solution of the overall skeleton. It can be seen from the principle that the more unknown skeletal points for iterative training, the better the fitting effect; however, at the same time, the amount of computation will also increase.

In order to obtain more ideal data, the skeletal points of each part need to be correlated. First, take the spine base point as the root node d and then the other skeletal points as the child nodes of the root node, similar to the tree model.

The child and parent node coordinates can be expressed in a matrix as follows:(6)$$M_{p,c}=T\left(t_{x},t_{y},t_{z}\right)\cdot R_{z}\left(\gamma \right)\cdot R_{y}\left(\beta \right)\cdot R_{x}\left(\alpha \right)$$
where $R_{z}\left(\gamma \right)$, $R_{y}\left(\beta \right)$, $R_{x}\left(\alpha \right)$ are(7)$$R_{z}\left(\gamma \right)=\left[\begin{array}{cccc} cos\gamma & -sin\gamma & 0 & 0\\ sin\gamma & cos\gamma & 0 & 0\\ 0 & 0 & 1 & 0\\ 0 & 0 & 0 & 1 \end{array}\right]$$(8)$$R_{y}\left(\beta \right)=\left[\begin{array}{cccc} cos\beta & 0 & sin\beta & 0\\ 0 & 1 & 0 & 0\\ -sin\beta & 0 & cos\beta & 0\\ 0 & 0 & 0 & 1 \end{array}\right]$$(9)$$R_{x}\left(\alpha \right)=\left[\begin{array}{cccc} 1 & 0 & 0 & 0\\ 0 & cos\alpha & -sin\alpha & 0\\ 0 & sin\alpha & cos\alpha & 0\\ 0 & 0 & 0 & 1 \end{array}\right]$$

$T\left(t_{x},t_{y},t_{z}\right)$ can be expressed as follows:(10)$$T\left(t_{x},t_{y},t_{z}\right)=\left[\begin{array}{cccc} 1 & 0 & 0 & t_{x}\\ 0 & 1 & 0 & t_{y}\\ 0 & 0 & 1 & t_{z}\\ 0 & 0 & 0 & 1 \end{array}\right]$$

Take any point $p={\left(x,y,z,1\right)}^{T}$ in the coordinate system of the child node, corresponding to $P={\left(X,Y,Z,1\right)}^{T}$ in the coordinate system of the parent node, and the relationship between them can be expressed as follows:(11)$$\left[\begin{array}{c} X\\ Y\\ Z\\ 1 \end{array}\right]=\left[\begin{array}{cccc} 1 & 0 & 0 & t_{x}\\ 0 & 1 & 0 & t_{y}\\ 0 & 0 & 1 & t_{z}\\ 0 & 0 & 0 & 1 \end{array}\right]\left[\begin{array}{cccc} R_{11} & R_{12} & R_{13} & 0\\ R_{21} & R_{22} & R_{23} & 0\\ R_{31} & R_{32} & R_{33} & 0\\ 0 & 0 & 0 & 1 \end{array}\right]\left[\begin{array}{c} x\\ y\\ z\\ 1 \end{array}\right]$$

The following Figure 3 shows the skeleton model represented by Euler angles. Some joints of the human body cannot rotate freely, so the value of the corresponding Euler angle is 0.

Each skeletal point can be represented by the root node and Euler angles. Taking the right hip as an example, it can be represented as follows:(12)$$P_{rh}=T\left(t_{x},t_{y},t_{z}\right)R_{0z}R_{0y}R_{0x}T_{rhr}\left[\begin{array}{c} 0\\ 0\\ 0\\ 1 \end{array}\right]$$

It is the same for the other positions.

### 3.2. LM Fitting Algorithm

The Levenberg–Marquardt (LM) algorithm is an efficient, fast, and widely used method for solving optimization problems involving objective functions. It is particularly effective in addressing least squares problems, offering advantages, such as rapid convergence, high stability, and a reduced likelihood of becoming trapped in local extrema.

The designed skeletal point fitting method mainly consists of the inverse kinematics constraint term E_IK_(θ, d), the smooth constraint term E_S_(θ, d) and the depth constraint term E_D_(θ, d), and the corresponding constraint coefficients are ω_IK_, ω_S_, and ω_D_. The overall constraint formula E(θ, d) can be expressed as:(13)$$E\left(\theta ,d\right)={\omega_{IK}E}_{IK}\left(\theta ,d\right)+{\omega_{S}E}_{S}\left(\theta ,d\right)+{\omega_{D}E}_{D}\left(\theta ,d\right)$$

Inverse kinematics was initially used mainly in robotic arm control and is now widely used in inverse kinematics for human posture assessment. In general, when operating on a data chain, the beginning part will be taken as input and the end part as output, while inverse kinematics is the opposite, which takes the end part as input and the beginning part as output. Its constraint formula is as follows:(14)$$E_{IK}\left(\theta ,d\right)={\left\|\left(P_{t}^{G}(\theta ,d)-d)-P_{t}^{L}(\theta ,d\right)\right\|}_{\;\;\;2}$$

To make the fitted skeletal point coordinates still time-stable, i.e., each frame before and after the skeletal point coordinates are correlated, a smoothing constraint needs to be added for optimization. This constraint is mainly optimized by controlling the speed of change of the connected frames before and after, and its constraint formula is as follows:(15)$$E_{s}\left(\theta ,d\right)={\left\|{\left(P_{t}^{G}(\theta ,d)\right)}^{n}-{\left(P_{t}^{G}\left(\theta ,d\right)\right)}^{n-1}/t\right\|}_{2}$$

To increase the stability of skeletal points, the optimization of 3D skeletal point coordinates requires constraints in the depth direction, that is, the z-axis direction of $P_{t}^{L}$. The depth constraint term optimizes the uncertainty in the z-axis direction of the coordinates by adjusting the acceleration with the constraint equation:(16)$$E_{d}\left(\theta ,d\right)={\left\|{{\left[((P_{t}^{G}(\theta ,d)\right)}^{n}-2{\left(P_{t}^{G}\left(\theta ,d\right)\right)}^{n-1}+{\left(P_{t}^{G}\left(\theta ,d\right)\right)}^{n-2})/t^{2}]}_{Z}\right\|}_{2}$$

The following steps have been designed to determine each constraint coefficient in $E\left(\theta ,d\right)$:

(i) Accurate and consecutive coordinate data of skeletal points in three frames must be selected as the output of the constraint algorithm. Then, the coordinate data of a certain skeletal point in the third frame must be changed as the input.

(ii) Set the values of the smoothing constraint term constraint coefficient $\omega_{s}$ and the depth constraint term constraint coefficient $\omega_{d}$ both to 1 and substitute the output and input parameters in the first step to evaluate the inverse kinematic constraint term constraint coefficient $\omega_{IK}$.

(iii) Set the value of $\omega_{d}$ to 1 and the value of $\omega_{IK}$ to the value derived in (ii), and substitute the parameters in (i) into the algorithm to determine the optimal solution of $\omega_{s}$;

(iv) Using the values of $\omega_{IK}$ and $\omega_{s}$ obtained in the previous steps and the value in (i) as parameters, the optimal solution of $\omega_{d}$ is obtained.

The above method is to be repeated to average the obtained parameters to obtain the final constraint coefficients.

Before using the LM algorithm to fit the constraint terms, the formula needs to be simplified, and the simplified formula is as follows:(17)$$E\left(\theta ,d\right)={\left\|e(\theta )\right\|}_{2}={\left\|{e(\theta )}^{\frac{1}{4}}\right\|}^{2}={\left\|f(\theta )\right\|}^{2}={f\left(\theta \right)}^{T}f\left(\theta \right)=F\left(\theta \right)$$

The corresponding Jacobian matrix is as follows:(18)$$J\left(\theta \right)=\frac{\partial f(\theta )}{\partial \theta }=\left[\begin{array}{cccc} \frac{\partial f_{1}(\theta )}{\partial \theta_{1}} & \frac{\partial f_{1}(\theta )}{\partial \theta_{2}} & \cdots & \frac{\partial f_{1}(\theta )}{\partial \theta_{n}}\\ \frac{\partial f_{2}(\theta )}{\partial \theta_{1}} & \frac{\partial f_{2}(\theta )}{\partial \theta_{2}} & \cdots & \frac{\partial f_{2}(\theta )}{\partial \theta_{n}}\\ \vdots & \vdots & \ddots & \vdots \\ \frac{\partial f_{m}(\theta )}{\partial \theta_{1}} & \frac{\partial f_{m}(\theta )}{\partial \theta_{2}} & \cdots & \frac{\partial f_{m}(\theta )}{\partial \theta_{n}} \end{array}\right]$$
where m and n are the number of rows and columns of the matrix, and their value is related to the number of unknown constraints.

The amount of change in the energy term is(19)$$F\left(\theta +h\right)=f\left(\theta \right)+j\left(\theta \right)h$$

The inverse of the energy term is(20)$$g\left(\theta \right)=\overset{´}{F\left(\theta \right)}=2{J_{\theta }}^{T}f_{\theta }$$

The amount of gradient change is(21)$$h=-\frac{g\left(\theta \right)}{\mu }.$$

Using the LM algorithm to fit and optimize the above formula, the steps are as follows:

(1) Complete the initialization of the parameters according to the above formula and calculate the value of $A={J_{\theta }}^{T}J_{\theta }$, where a_ii_ is the value of the i-th row and i-th column in matrix A.

(2) Set the allowed differences ε, ε_2_, the coefficients v, the damping factor μ_0_, and the maximum number of iterations K_max_ that can be made so that k = 0 and μ = τ*max{a_ii_} (max{a_ii_} denotes the maximum value between all data in the matrix).

(3) Compare the value of ${\left\|g(\theta )\right\|}_{\infty }$ with ε. If the difference between the two is less than 0, stop the iteration and output the current value of θ and d; otherwise, continue the next iteration.

(4) k = k + 1, if the value of k at this point is greater than K_max_ then end the iteration; otherwise, calculate the value of h at this point. If $h\leq \varepsilon_{2}(\left\|\theta \right\|+\varepsilon_{2})$, output the result and end this iteration; otherwise, go to the next iteration.

(5) θ_new_ = θ + h, coefficient ρ = (F(θ) − F(θ_new_))/h^T^(μh − g(θ_new_)), if ρ > 0, then go to step (6), otherwise go to step (7).

(6) Complete the reassignment of θ = θ_new_, $A={J_{\theta }}^{T}J_{\theta }$, $g\left(\theta \right)=2{J_{\theta }}^{T}f_{\theta }$, $\mu =\mu \times max\{\frac{1}{3},1-{\left(2\rho -1\right)}^{3}\}$ and v = 2 and return to step (3).

(7) Complete the re-assignment of μ = μ × V and V = 2V and return to step (3).

The final result $P_{t}^{G}(\theta ,d)$ is obtained by the above iterations.

Where the correlation coefficients involved in the iterative process are set as follows: ε = 10^−12^, ε_2_ = 10^−12^, v = 2, K_max_ = 100, and τ = 10^−6^.

The stability of the results is improved by defining the necessary constraints and constraint coefficients for the LM algorithm. The algorithm was designed to take two 3D skeletal point coordinates as input, as opposed to one 2D coordinate and one 3D coordinate. This approach eliminated the need to incorporate projection constraint terms into the algorithm’s constraints, thereby reducing the computational load of the LM algorithm. In addition, the constraint coefficients were optimized to improve the algorithm efficiency.

## 4. Algorithm Testing and Analysis

The experimental setup consisted of a Kinect V2 sensor and a PC equipped with 16GB RAM, an Intel i7-7700 processor, an NVIDIA GTX 1060 6G graphics card (the above hardware is branded as Lenovo and manufactured in Beijing, China.), and the Windows 10 operating system.

To address the issues of jitter and self-occlusion in the Kinect algorithm, as well as the susceptibility of the OpenPose algorithm to complex backgrounds and environmental interference, experiments were conducted in three different scenarios to validate the effectiveness of the proposed algorithm. The experiments compared the human skeletal points extracted by the three algorithms, and their recognition rates and processing speeds were subsequently evaluated.

### 4.1. Qualitative Analysis of Experimental Results

Three different scenarios were tested using the OpenPose algorithm, Kinect V2’s algorithm, and the proposed algorithm in this paper. The OpenPose algorithm can obtain an 18 2D skeletal points sequence of the human body, Kinect V2’s own algorithm can obtain a 25 3D skeletal points sequence, and the proposed algorithm in this paper can obtain a 15 3D skeletal points sequence. The details of the test environment and results are shown as follows.

Scene 1 is a complex background with a large number of elements. Among them, the target human body exists in the first frame, the 13th frame image was randomly selected, and the experimental results are shown in Figure 4. It can be seen that the algorithm can correctly obtain the skeletal points of the human body in a complex context. The Kinect algorithm has some errors in the feet due to distance and angle issues, and the OpenPose algorithm also has recognition errors. The OpenPose algorithm incorrectly identifies humanoid objects or images that appear in the background as human bodies, and the distance and actions of the experimenter in some scenes will have some effect on the skeletal points.

Scene 2 is a low illumination situation. Among them, there is no complete target human body in the first frame, the 17th frame image was randomly selected, and the experimental results are shown in Figure 5. Under low illumination conditions, the OpenPose algorithm cannot accurately capture human body information and exhibits missing and offset phenomena. In contrast, the other two algorithms successfully extract correct skeletal point information. In complete darkness, OpenPose cannot recognize the human body at all. These results demonstrate that both the Kinect algorithm and the proposed algorithm in this study are robust to changes in illumination.

Scene 3 is a situation with partial self-occlusion of the human body. Among them, the target human body exists in the first frame, the ninth frame image was randomly selected, and the experimental results are shown in Figure 6. It can be seen that the skeletal points acquired by Kinect will be jittery or even wrong when the target person is in a self-occlusion situation, while the other two algorithms can correctly acquire the human skeletal points by algorithmic prediction. When the occlusion phenomenon is serious, all three algorithms will show errors, such as partial loss of skeletal points.

### 4.2. Human Action Analysis Based on DTW Algorithm

During the process of action recognition, even if an action is performed as accurately as possible, discrepancies may still arise between the observed action sequence and the standard action sequence along the time axis. This is because individuals perform actions at varying speeds, and there is no straightforward frame rate correspondence between the actions of different individuals and the standard action. By employing the DTW algorithm, the two sequences can be effectively aligned, thereby enhancing the accuracy of action recognition. Furthermore, the greater the deviation observed during the alignment process, the less standardized the performed action is considered to be.

Since the time series of 3D skeletal point coordinates may be affected by different individuals, the time series of skeletal point coordinates is first converted into the time series of skeletal point angles, and then compared with the standard series. The application of the DTW algorithm in action recognition is described by the simple action of “right shoulder abduction and elbow flexion”. The action of “right shoulder abduction and elbow flexion” is that the right hand is lifted from the natural hanging state to the horizontal state of 180 degrees, and then the big arm is kept stable, the right elbow is bent, and the small arm and the big arm of the right arm are in the shape of a right angle, and then the natural hanging of the arm is restored, as shown in Figure 7. This action mainly uses the time series of the angles of the two skeletal points of the right shoulder and the right elbow as the action sequence for standard action estimation.

Let $P_{rb},P_{rs},P_{re},P_{rw}$ be the 3D spatial coordinate points of the right thoracic spine point, right shoulder, right elbow, and right wrist, respectively, and $C_{rs},C_{re}$ be the angle of the right shoulder skeletal point and the angle of the right elbow skeletal point, respectively. Then $C_{rs},C_{re}$ can be obtained by Equations (22) and (23):(22)$$C_{rs}=arccos\frac{\left(P_{rs}-P_{rb})\cdot (P_{rs}-P_{re}\right)}{\left|P_{rs}-P_{rb}\right|\times \left|P_{rs}-P_{re}\right|}$$(23)$$C_{re}=arccos\frac{\left(P_{re}-P_{rs})\cdot (P_{re}-P_{rw}\right)}{\left|P_{re}-P_{rs}\right|\times \left|P_{re}-P_{rw}\right|}$$

The 3D skeletal point coordinate time series of the right thoracic spine point, right shoulder, right elbow, and right wrist are obtained and then converted into the right shoulder skeletal point angle time series $X_{rs}$ and the right elbow skeletal point angle time series $X_{re}$ by the two equations mentioned above. According to the DTW algorithm, the path distance between the experimentalist’s right shoulder skeletal point time series $X_{ls}$ and the standard action’s right shoulder skeletal point time series $Y_{rs}$ is calculated as $D_{rs}$, and the path length is $K_{rs}$.

The DTW distance between the experimenter’s right elbow skeleton point time series $X_{re}$ and the standard action right elbow skeleton point time series $Y_{re}$ is $D_{re}$, and the path length is $K_{re}$. Let the weight of the right shoulder skeleton point in the action evaluation be $w_{rs}$ and the weight of the right elbow skeleton point in the action recognition be $w_{re}$. The DTW similarity ${DTW}_{value}$ of the “right shoulder adduction elbow flexion” action can be obtained by Equation (24):(24)$$\left\{\begin{array}{l} w_{rs}\times \frac{D_{rs}}{K_{rs}}+w_{re}\times \frac{D_{re}}{K_{re}}={DTW}_{value}\\ w_{rs}+w_{re}=1\;\;\;\;\;\;\;\;\;\;\;\;\;\;\;\;\;\;\;\;\;\;\;\;\;\;\;\;\;\;\;\;\;\;\;\;\;\;\;\;\;\;\; \end{array}\right.\;\;$$

If a threshold value λ is selected, when ${DTW}_{value}\leq \lambda$, the more standard the experimenter’s “right shoulder abduction elbow bend” movement is completed; otherwise, it means the experimenter’s movement is not standard enough and needs to be adjusted. Other movements are similar to it and will not be described.

To validate the synergy between the LM algorithm and the DTW algorithm, we verified the effectiveness of the algorithm through ablation experiments. The dataset used is Human 3.6M, which is the dataset with the largest number of samples and the most widely used in the 3D domain. The dataset contains about 3.6 million 3D human poses and corresponding images with 11 experimenters and 17 action scenes. The results of the ablation experiments are shown in Table 2.

The results show that the LM+DTW co-design scheme significantly improves the action recognition accuracy (MPJPE = 49.9 mm) and robustness (anomalous frame rate of 4.8%), which is better than the other three schemes. Although not all potential sources of error (e.g., sensor noise) are covered, future work will further minimize the discrepancy by repeating the experiments and tighter definitions of anomalous frames.

### 4.3. Quantitative Analysis of Experimental Results

To evaluate the accuracy of the algorithm’s recognition capabilities, five actions were selected: shoulder lateral press, shoulder press, Arnold press, deep squat, and deadlift. These five movements were chosen because they engage nearly all parts of the body, enabling the detection of joint information required for the study. Thirty volunteers were invited to participate in the action recognition test. Each volunteer was asked to perform the five actions separately, repeating each action 20 times. This process generated a total of 600 datasets, which were used to observe the number of correct recognitions.

Firstly, action recognition was performed using only the Kinect algorithm, and the test results are presented in Table 3.

Secondly, the proposed algorithm in this study was tested for partial complex action recognition, and the results are presented in Table 4.

The experimental results demonstrate that the recognition rate of the proposed algorithm is effectively improved over the original Kinect algorithm. The recognition rate is not affected by the volunteer’s body size, with good robustness.

Finally, a t-test was executed to provide additional validation of the method’s accuracy. The t-test results, as presented in Table 5, demonstrate the efficacy of the algorithm optimization, with a statistical significance level of *p* < 0.01, indicating that the enhancement in action recognition rates exceeds the statistical threshold. The mean improvement observed was 5.02 percentage points.

The results, such as confidence intervals and standard deviations, are shown in Table 6.

(1) Standard deviation calculation: Based on the binomial distribution assumption (each action is independent and obeys the Bernoulli distribution), the formula is σ = *p*(1 − p)n, where n = 600 (30 people × 20 reps/person).

(2) Confidence interval: Calculated using normal approximation, range *p* ± 1.96σ.

(3) Significance test: All actions were significantly better than random guessing by a one-sample z-test (baseline accuracy assumed to be 50%) (*p* < 0.001).

(4) ANOVA: The difference between the groups was significant (F = 65.2, *p* < 0.001), indicating that there was a statistical difference in the difficulty of recognizing different movements. Tukey HSD showed that the deep squat was significantly less accurate than the other movements due to the susceptibility to postural bias due to the synergy of multiple joints in the body (hips, knees, and ankles) (*p* < 0.01). The shoulder lateral press was not significantly different from the shoulder press due to the similarity of the trajectory of the movements (both involving shoulder flexion, shoulder abduction, and elbow extension) (*p* = 0.12).

(5) Multiple testing correction: The primary results remained stable after Bonferroni correction (α = 0.005 after correction).

Overall, the average recognition rate of all actions is 95.94%, the total standard deviation σ = 3.2%, and the confidence interval is [92.7%, 99.2%]. The model recognition performance is stable, the confidence interval coverage is reasonable, and the standard deviation verifies low volatility, which can support subsequent action classification optimization and statistical inference.

The motion recognition capabilities demonstrated in this experiment can be used to simulate robot interaction scenarios. In this regard, we designed and implemented an avatar interaction system to enable interaction between humans and virtual robots through data transfer between Unity3D and Kinect. It allows users to learn specified fitness movements and interact virtually. The interaction interface is shown in Figure 8. The system received positive feedback and demonstrated effectiveness.

During the detection process, we found some issues, such as noticeable errors and even loss of skeletal points. These errors were determined to be caused by the distance and angle between the human body and the Kinect sensor. To investigate further, additional tests were conducted.

The first test focused on distance. The human body was positioned upright, while the Kinect sensor was placed horizontally at a height adjusted to match the human body. Each distance was tested 300 times, and the results are presented in Table 7.

Next, the test was performed by adjusting the angle between the body and the Kinect sensor at a fixed distance, following the same steps as above. The results are shown in Table 8.

It is evident that when performing human body recognition, the distance, angle, and relative positioning between the human body and the Kinect sensor must be carefully adjusted. Failure to do so may result in specific issues, such as recognition errors.

Finally, the speed of skeletal point recognition of the three algorithms was tested several times, and the results are shown in Table 9.

It can be seen that compared to the Kinect algorithm, the optimized algorithm has improved skeletal point prediction, higher stability and accuracy, and better performance in recognizing complex movements. Additionally, the average recognition speed is slightly improved compared to the standalone OpenPose algorithm.

## 5. Conclusions and Future Works

### 5.1. Conclusions

A multi-algorithm co-optimization approach is proposed in this paper, which combines the methods of Kinect and OpenPose algorithms to optimize the recognition rate by using the LM algorithm and to improve the complexity and real-time performance of action recognition by using the DTW algorithm. The Kinect and OpenPose algorithms are used to acquire skeletal points, and input data are obtained by processing human skeletal points through operations, such as mapping and translation. Subsequently, the LM algorithm is applied to fit the input data and complete data fusion, resulting in a stable 3D skeletal point sequence. Finally, the DTW algorithm is integrated for action estimation. The proposed method is robust to environmental factors, such as illumination and complex backgrounds. It effectively addresses issues, such as jitter and self-occlusion, encountered when using the Kinect algorithm to obtain skeletal points. The skeletal point recognition performance is stable, with an average recognition rate of 95.94% for the designed complex actions, and the optimized algorithm also demonstrates improved processing speed.

### 5.2. Future Works

Human–robot interaction is a rapidly developing research field with great potential for future development, which could be applied to many fields, such as bionic robotics. While the proposed method demonstrates promising results, several challenges and practical limitations must be acknowledged to guide future work.

Current limitations:

(1) Sensitivity to noise, where the method’s performance may degrade under high noise levels, such as motion blur, sensor inaccuracies, or environmental disturbances (e.g., reflective surfaces generating point cloud noise).

(2) Fixed camera assumptions, which require the user to be always within 1.5–3 m of the line-of-sight, and the recognition failure rate increases dramatically when lateral movement exceeds 30°, which limits its applicability in dynamic scenes.

(3) The current validation dataset is also limited, and the number of designed actions has to be increased. No experimental validation has been conducted for types, such as older adults or children.

(4) Due to some complexity in algorithmic synergy, there is still room for optimizing efficiency.

Future works:

(1) Improving noise robustness could be achieved by developing multi-sensor confidence-weighted fusion or introducing filtering methods.

(2) For distance and range limitations, the latest Azure Kinect could be used instead of Kinect V2 to improve the detection range and depth. Meanwhile, in-depth research could be conducted for mobile scene adaptation.

(3) A broader population and more exercise designs could be included, which would help to identify and address potential unidentified problems.

(4) The complexity of the algorithm could be optimized by further delving into the LM fitting process and the synergistic process to increase the speed. The recognition rate could be further improved by combining it with the Particle Swarm Optimization algorithm.

By addressing these challenges, future work could bridge the gap between experimentation and real-world deployment, especially in human–robot interaction, bionic robotics, etc. Collaborative efforts across algorithm design, sensor technology, and energy-efficient computing will be critical to unlocking scalable solutions.

## Figures and Tables

**Figure 1 biomimetics-10-00254-f001:**
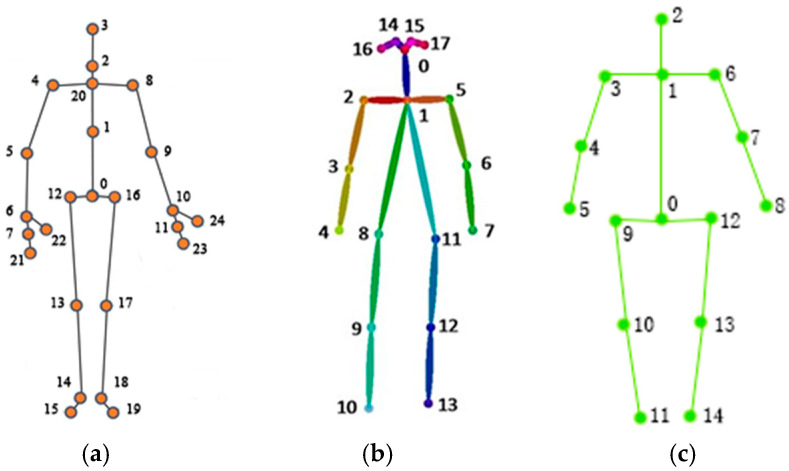
Skeletal map obtained by Kinect, OpenPose, and the algorithm in this paper. (**a**) Kinect. (**b**) OpenPose. (**c**) Skeletal points of this article.

**Figure 2 biomimetics-10-00254-f002:**
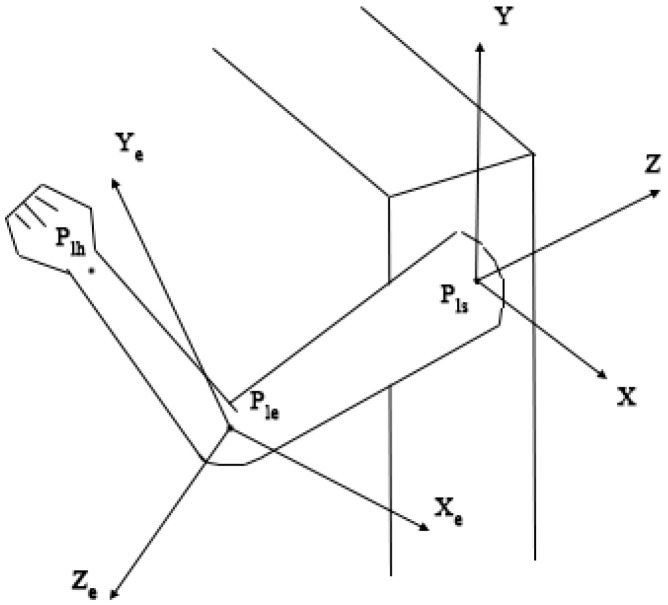
Elbow coordinate system.

**Figure 3 biomimetics-10-00254-f003:**
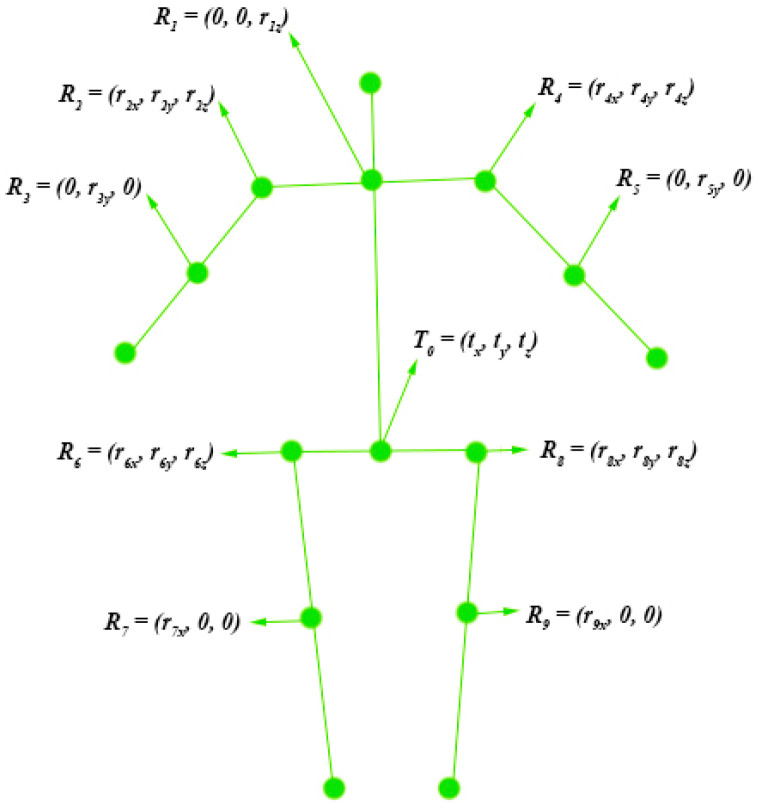
Human skeleton model represented by Euler angles.

**Figure 4 biomimetics-10-00254-f004:**
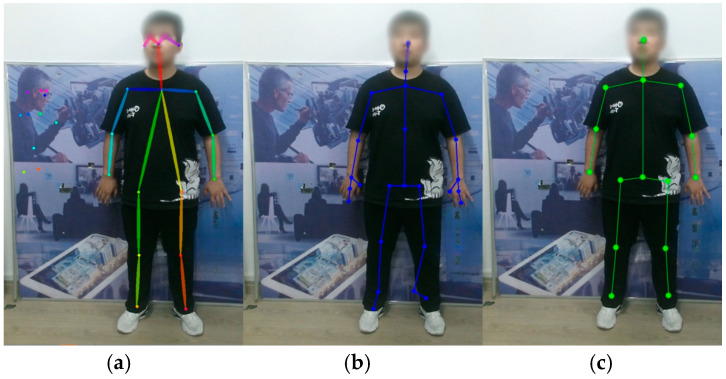
Detection results in a complex background. (**a**) OpenPose algorithm. (**b**) Kinect algorithm. (**c**) Algorithm in this paper.

**Figure 5 biomimetics-10-00254-f005:**
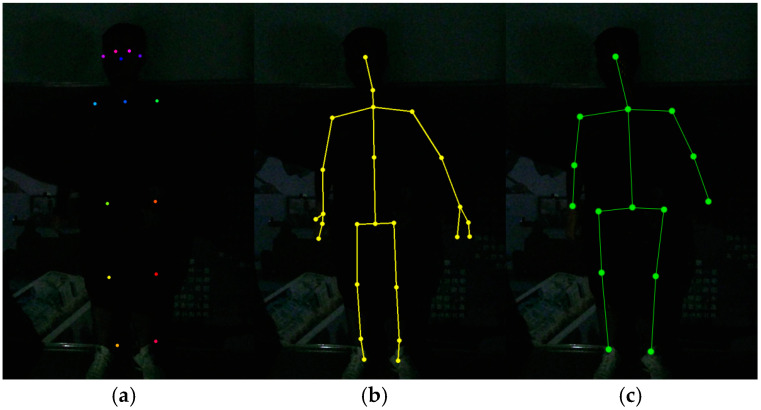
Detection results at low illumination. (**a**) OpenPose algorithm. (**b**) Kinect algorithm. (**c**) Algorithm in this paper.

**Figure 6 biomimetics-10-00254-f006:**
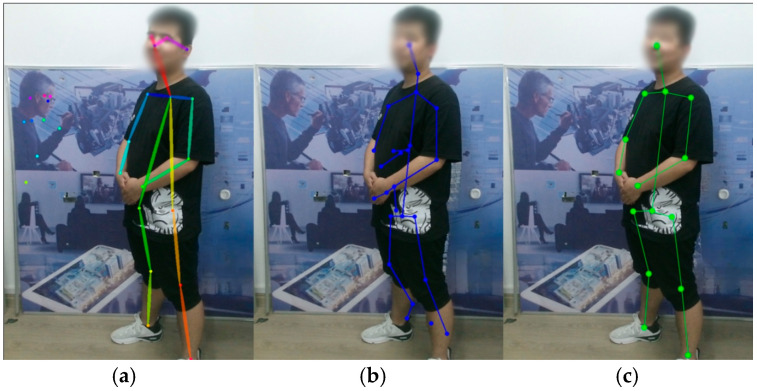
Detection results when self-occlusion occurs. (**a**) OpenPose algorithm. (**b**) Kinect algorithm. (**c**) Algorithm in this paper.

**Figure 7 biomimetics-10-00254-f007:**
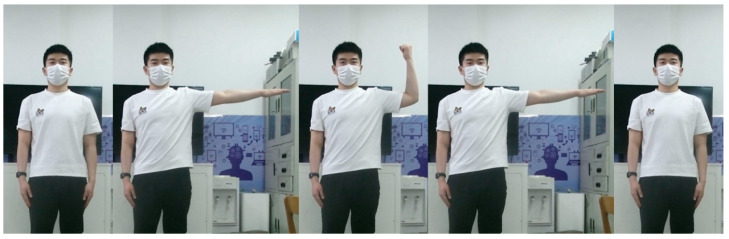
Schematic diagram of right shoulder abduction and elbow flexion movement.

**Figure 8 biomimetics-10-00254-f008:**
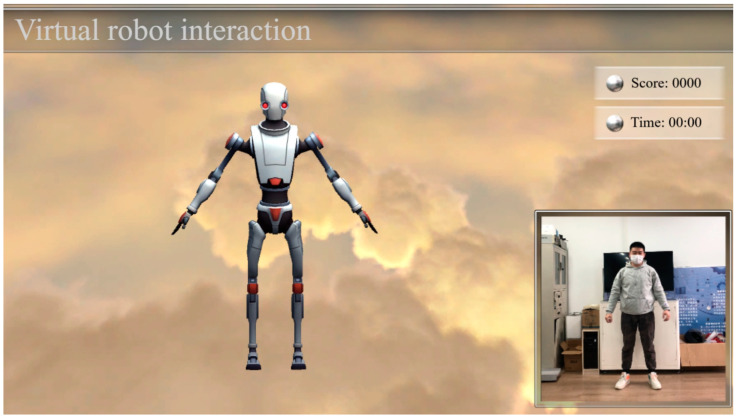
Virtual robot interaction interface.

**Table 1 biomimetics-10-00254-t001:** Multi-dimensional comparison.

Dimensions	Traditional ML Method	Deep Learning Method	Ours
Data Modal Fusion	Unimodal (RGB/Skeleton)	Multimodal independent processing	Multimodal dynamic alignment
Feature Extraction	Manual design of time-domain statistics	Automatic extraction but spatiotemporal coupling	DTW-driven decoupling mechanism for spatiotemporal features
Labeling Cost	Manual frame-by-frame labeling	Requires 10w+ labeled samples	Zero labeling
Optimization mechanism	Static hyperparameter	Global gradient descent	Multi-stage optimization of LM guidance
Occlusion Robustness	Data Enhancement Compensation	Eigenspace interpolation	Multi-source complementarity

**Table 2 biomimetics-10-00254-t002:** The results of ablation experiments.

Schemes	MPJPE (mm)	DTW Cumulative Deviation	Abnormal Frame Rate (%)
LM + DTW	49.8	1.27	4.8
LM only	68.7	3.15	12.5
DTW only	61.2	2.84	9.1
LM + Euclidean	57.8	2.09	6.7

**Table 3 biomimetics-10-00254-t003:** Kinect algorithm test table.

Action	Number of People	Number of Actions	Number of Correct Recognitions	Recognition Rate (%)
shoulder lateral press	30	20	578	96.3
shoulder press	30	20	570	95
Arnold press	30	20	522	87
deep squat	30	20	512	85.3
deadlift	30	20	546	91

**Table 4 biomimetics-10-00254-t004:** Algorithm test table.

Action	Number of People	Number of Actions	Number of Correct Recognitions	Recognition Rate (%)
shoulder lateral press	30	20	594	99
shoulder press	30	20	592	98.7
Arnold press	30	20	566	94.3
deep squat	30	20	550	91.7
deadlift	30	20	576	96

**Table 5 biomimetics-10-00254-t005:** *t*-test results.

Action	Kinect Algorithm	Ours	z Value	*p* Value	Significant or Not
shoulder lateral press	96.3%	99.0%	−3.52	0.0004	yes
shoulder press	95.0%	98.7%	−3.94	<0.001	yes
Arnold press	87.0%	94.3%	−5.11	<0.001	yes
deep squat	85.3%	91.7%	−4.22	<0.001	yes
deadlift	91.0%	96.0%	−3.71	0.0002	yes

**Table 6 biomimetics-10-00254-t006:** SD and 95% CI results.

Action	Accuracy (%)	SD(σ)	95% CI	*p* Value
shoulder lateral press	99.0%	±0.41	[98.6%, 99.4%]	<0.001
shoulder press	98.7%	±0.43	[98.3%, 99.1%]	<0.001
Arnold press	94.3%	±0.93	[93.4%, 95.2%]	<0.001
deep squat	91.7%	±1.12	[90.6%, 92.8%]	<0.001
deadlift	96.0%	±0.78	[95.2%, 96.8%]	<0.001

**Table 7 biomimetics-10-00254-t007:** Distance test of the human body and Kinect.

Distance (m)	Kinect Algorithm	Recognition Rate (%)	Ours	Recognition Rate (%)
1.5	189	63	213	71
2.0	278	92.67	286	95.33
3.0	300	100	300	100
3.5	286	95.33	300	100
4.0	254	84.67	273	91
4.5	102	34	125	41.67

**Table 8 biomimetics-10-00254-t008:** Angel test of the human body and Kinect.

Angel (°)	Kinect Algorithm	Recognition Rate (%)	Ours	Recognition Rate (%)
0	300	100	300	100
30	289	96.33	300	100
45	256	85.33	287	95.67
60	213	71	264	88
90	0	0	0	0
0	300	100	300	100

**Table 9 biomimetics-10-00254-t009:** Recognition speed comparison table.

No.	Kinect Recognition Speed (s)	OpenPose Recognition Speed (s)	Ours (s)
1	0.57612	1.11011	1.04361
2	0.60053	1.09477	1.01948
3	0.59206	1.08447	1.03209
4	0.54681	1.13548	1.04618
5	0.61435	1.22148	1.09143
6	0.55164	1.30686	1.20437
7	0.59631	1.24357	1.03421
8	0.60321	1.04312	1.08437
9	0.57619	1.08342	1.07681
10	0.60746	1.13465	1.12648

## Data Availability

The original contributions presented in this study are included in the article. Further inquiries can be directed to the corresponding author.

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
