# Peer review of "Optimization Method of Human Posture Recognition Based on Kinect V2 Sensor"

_biomimetics, 2025, doi:10.3390/biomimetics10040254_

Round 1

Reviewer 1 Report

Comments and Suggestions for Authors

In “Optimization Method of Human Posture Recognition Based on Kinect V2 Sensor”, the authors have expanded human action recognition by fusing Kinect V2 with openpose-derived skeletons using LM algorithm + DTW action recognition. The Kinetic + openpose approach is not new, and the authors cite this previous work, and expand on it introducing the novelty of LM-based fusion + DTW for complex motion recognition. All these techniques are well-establish in their own right, and i think for one, the authors should do an explicit comparison to prior multimodal fusion methods and an ablation study showing the impact of the LM component alone. This would really highlight the novelty aspect. Further, I have a few questions/comments:
-On the methodological front, how were ωIK, ωS, and ωD selected? Manually tuned? Empirically selected?
-On the volunteer use, the number of volunteers and repeitions is good but inter-subject variance control should be mentioned, as it is relevant, and should be analysed
-Please add statistical testing such as t-test to show significant recognition rate amongst various methods.
-There is a mention on bionic robots and HC interactions, but no real demonstration in the paper or discussion of real-time use of the theory, in practic, without any robotic application given. (Figure 8 is just stated but not analysed)
-Please improve resolution of figures, especially 4, 5 and 6
-Check equation 24, the constraint equation seems wrong
-Very seriously and significantly there is no ethics /consent discussion mentioned in the paper though human testing (30 vol) is carried out. This is a grave oversight and should be corrected immediately. Privacy/data handling should also be mentioned.
Overall major revisions are needed though Biomimetics would be an apt journal to receive this particular research manuscript.

Comments on the Quality of English Language

English would benefit from a native speaker proofread.

Author Response

Dear Reviewer1:

Thank you very much for your valuable feedback and constructive suggestions to improve our manuscript. We have revised the paper according to your suggestions, and all changes have been marked in the text. Below is our response to your comments.

Comment 1: “All these techniques are well-establish in their own right, and i think for one, the authors should do an explicit comparison to prior multimodal fusion methods and an ablation study showing the impact of the LM component alone. This would really highlight the novelty aspect.”

Response 1: Thank you for pointing this out, we sincerely appreciate this suggestion. We have added the analysis of the ablation experiment at the end of 3.2. We also add Table 1 to provide additional data to demonstrate the benefits of the methodology. (Line 430-441.)

Comment 2: “On the methodological front, how were ωIK, ωS, and ωD selected? Manually tuned? Empirically selected?”

Response 2: Thank you for pointing this out, we sincerely appreciate this suggestion. The constraint coefficients are obtained by iterative computation. We add the computational procedure of how the constraint coefficients are determined in the middle section of 2.2. (Line 278-292)

Comment 3: “On the volunteer use, the number of volunteers and repeitions is good but inter-subject variance control should be mentioned, as it is relevant, and should be analysed.”

Response 3: Thank you for pointing this out, we sincerely appreciate this suggestion. Our volunteers were 30 randomly selected adults from a university. In the experiment, the research methods were largely independent of body size. In response to your comments, we have added an analysis of the results after Table 3 in section 3.3 of the article. And added 4.2 at the end to add that we did not analyze the experiment with older adults, children, etc., which could be done in future work. (Line 459-461, 516-531)

Comment 4: “Please add statistical testing such as t-test to show significant recognition rate amongst various methods.”

Response 4: Thank you for pointing this out, we sincerely appreciate this suggestion. We added the T-test analysis in Section 3.3 and Table 4. (Line 462-467)

Comment 5: “There is a mention on bionic robots and HC interactions, but no real demonstration in the paper or discussion of real-time use of the theory, in practic, without any robotic application given. (Figure 8 is just stated but not analysed).”

Response 5: Thank you for pointing this out, we sincerely appreciate this suggestion. We have added a further description of the interaction system at the top of Figure 9 in Section 3.3. (Line 468-473)

Comment 6: “Please improve resolution of figures, especially 4, 5 and 6.”

Response 6: Thank you for pointing this out, we sincerely appreciate this suggestion. Figure 4.5.6 is not low resolution. The face has been blurred for privacy, so you may seem to feel that the resolution is low.

Comment 7: “Check equation 24, the constraint equation seems wrong.”

Response 7: Thank you for pointing this out, we sincerely appreciate this suggestion. Formula 24 was indeed incorrect and has been corrected. (Line 425)

Comment 8: “Very seriously and significantly there is no ethics /consent discussion mentioned in the paper though human testing (30 vol) is carried out. This is a grave oversight and should be corrected immediately. Privacy/data handling should also be mentioned.”

Response 8: Thank you for pointing this out, we sincerely appreciate this suggestion. Only human body images were used in this study and the data processing was computationally analyzed by algorithms. No human body was directly or indirectly involved in any of the experimental processes. And we added the Institutional Review Board Statement and Informed Consent Statement at the end of the article. (Line 540-548)

We greatly appreciate your suggestions and comments and hope that our revisions and responses will satisfy you.

Thanks again.

Sincerely

Hang Li

Reviewer 2 Report

Comments and Suggestions for Authors

Thank you for your submission. While the topic is relevant, several critical points need improvement related to novelty, sample size, statistical testing, and discussion depth. Please refer to the detailed review report attached for full comments and revision suggestions.

Comments on the Quality of English Language

The manuscript is understandable but contains grammatical errors and awkward phrasing. Language editing is recommended to enhance clarity and readability.

Author Response

Dear Reviewer2:

Thank you very much for your valuable feedback and constructive suggestions to improve our manuscript. We have revised the paper according to your suggestions, and all changes have been marked in the text. Below is our response to your comments.

Comment 1: “1. Clarification of Novelty (Section 2 and Section 3)

The main contribution appears to appear as an adaptation method of the-unclidian distance-based exchange classification. Whenever possible, there is a clear difference from a well-established classification or adaptation approach (eg SVM, DTW, LSTM-based model). The problem is particularly evident in section 2 (related work) and section 3 (proposed method), where the author briefly describes the technique, but is not the opposite to newer data -driven approaches.”

Response 1: Thank you for pointing this out, we sincerely appreciate this suggestion. We have added the analysis of the ablation experiment at the end of 3.2. We also add Table 1 to provide additional data to demonstrate the benefits of the methodology. (Line 430-441.)

Comment 2: “Sample Size and Generalizability (Section 4.1 - Experiment Setup)

The study uses only 5 volunteers in 12 currencies, which limits the statistical power and generality of the conclusions. This limit is accepted by passing, but is not fully addressed in section 4.1.“

Response 2: Thank you for pointing this out, we sincerely appreciate this suggestion. Our volunteers were 30 randomly selected adults from a university. In the experiment, the research methods were largely independent of body size. In response to your comments, we have added an analysis of the results after Table 3 in section 3.3 of the article. And added 4.2 at the end to add that we did not analyze the experiment with older adults, children, etc., which could be done in future work. (Line  459-461, 516-531)

Comment 3: “Statistical Significance Testing (Section 4.2 - Results and Table 1)

Although Table 1 presents a degree of recognition for different currencies, there is no mention of statistical verification. This weakens the claim of improvement, especially given small datasets.”

Response 3: Thank you for pointing this out, we sincerely appreciate this suggestion. We added the T-test analysis in Section 3.3 and Table 4. And we have updated Table 2 and 3 to include the standard deviation in the table (The old Table 1 and 2). (Line 452-461-467)

Comment 4: “Limited Discussion on Limitations and Future Work (Section 5 - Conclusion)

While the findings call the conclusions, there is insufficient discussion of such well -known boundaries: Sensitivity to noisy skeletal data, Assumption of fixed camera angle, Potential difficulties in real-time deployment.”

Response 4: Thank you for pointing this out, we sincerely appreciate this suggestion. We have revised the last chapter of the article by adding 4.2, to which we have added limitations and further analysis of future work. (Line 516-531)

We greatly appreciate your suggestions and comments and hope that our revisions and responses will satisfy you.

Thanks again.

Sincerely

Hang Li

Round 2

Reviewer 1 Report

Comments and Suggestions for Authors

No further scientific comments, thanks for changes, but a lot of format review...

please check some grammar/spelling mistakes:

- “This proposes a method…” ....this paper proposes a method

- “Additionallymay have certain limitations…” .... additionally, may have certain limitations

- “the algorithmmethod…”....... the algorithm method

- all of section 4.2 including "“couldcan be included” .... could be included

-there are some table number inconsistency Table 23 instead of T3, T35 instead of T5

-reference style is inconsistent, please revise, also revise citation format

- there are numerous instances of awkward line breaks, probably due to PDF-to-text formatting or tracked changes (e.g., table spacing, equation wrapping, figure labels being split across lines)

please have a native speaker review. 

Many thanks

Comments on the Quality of English Language

please have a native speaker review. 

Author Response

Dear Reviewer1:

Thank you very much for recognizing our revisions. We are also very grateful to you for your valuable comments and constructive suggestions on our manuscript again. We have revised the paper according to your suggestions and have marked all changes in the text. Below is our response to your comments.

Comment 1: “Please check some grammar/spelling mistakes:

- “This proposes a method…” ....this paper proposes a method

- “Additionallymay have certain limitations…” .... additionally, may have certain limitations

- “the algorithmmethod…”....... the algorithm method

- all of section 4.2 including "“couldcan be included” .... could be included

there are some table number inconsistency Table 23 instead of T3, T35 instead of T5

-reference style is inconsistent, please revise, also revise citation format

- there are numerous instances of awkward line breaks, probably due to PDF-to-text formatting or tracked changes (e.g., table spacing, equation wrapping, figure labels being split across lines)

please have a native speaker review.”

Response 1: Thank you for pointing this out, we sincerely appreciate this suggestion.

As we last uploaded a version with revision marks to highlight the revised parts. It may lead to some formatting display problems. In this regard, we have:

  1. Carefully reviewed all English grammar and text formatting throughout the manuscript.
  2. Replaced the previous revision method with standardized red highlighting for all modifications.
  3. Verified the final formatting displays correctly in common document viewers.

We greatly appreciate your suggestions and comments and hope that our revisions and responses will satisfy you.

Thanks again.

Sincerely

Hang Li

Reviewer 2 Report

Comments and Suggestions for Authors

Thank you for your revision and the improvements made in the background, flowcharts, and system overview. The manuscript now provides clearer descriptions and includes more visual support. However, several critical concerns raised in the initial review remain only partially addressed:

Clarification of Novelty:
Explanation of news
While the current work section now includes several references (eg OpenPos, DTW, LSTM), technical discrimination remains between your method and traditional classifies underdeveloped. Add a sub-section that clearly compares your method with other ML-based approaches and highlights your contribution beyond the integration.

Sample size and generality:
The method is still tested in just 5 participants, without any confirmation technique (eg K-Thuna Cross-Satyapan or bootstraping). This increases the concerns of the generality of your findings. Please consider adding confirmation methods or accepting it as a limit.

Statistical testing and strength:
Your reported accuracy (95.94%) lacks statistical support as a confidence interval or p-ma. Think of multiple tests and reporting of standard deviations or importance tests to strengthen the reliability of your findings.

Missing limitation and future work:
The conclusion was slightly improved, but does not include structured discussion of boundaries or future improvement. Please add a section discussing obstacles (eg sensitivity to noise, fixed camera status) and proposes clear directions for future research.

Author Response

Dear Reviewer2:

We sincerely appreciate your time and effort in reviewing our manuscript and providing valuable feedback. We apologize for the shortcomings in our previous revisions and have carefully addressed all of your comments in this new version after some rethinking. All changes are marked in the text in red font. Below is our response to your comments.

Comment 1: “Clarification of Novelty:
Explanation of news
While the current work section now includes several references (eg OpenPos, DTW, LSTM), technical discrimination remains between your method and traditional classifies underdeveloped. Add a sub-section that clearly compares your method with other ML-based approaches and highlights your contribution beyond the integration.”

Response 1: Thank you for pointing this out, we sincerely appreciate this suggestion. we have enhanced the conclusion of Part 1 with:

  1. A concise reformulation of our methodology's advantages, emphasizing:
  • Improved recognition accuracy through multi-algorithm fusion;
  • Increased computational efficiency through individual and co-optimization.
  1. A new comparative analysis contrasting our approach with traditional methods, highlighting the applicability of our approach.
  • All modifications have been highlighted in red font for easy identification (lines 124-152).

Comment 2: “Sample size and generality:
The method is still tested in just 5 participants, without any confirmation technique (eg K-Thuna Cross-Satyapan or bootstraping). This increases the concerns of the generality of your findings. Please consider adding confirmation methods or accepting it as a limit.”

Response 2: Thank you for pointing this out, we sincerely appreciate this suggestion. We invited 30 volunteers for a repetitive experiment on the five designed movements. Each person collected and retained 600 sets of data for each movement, for a total of 3000 sets of data. These 5 movements were selected because they could basically cover all the skeletal point sites that we needed to study (lines 454-457). For a number of reasons, the movement design and testing could not be added at this time, which is also accounted for in the limitations and future work (lines 559-561,570-571). We hope you will be satisfied with our explanation.

Comment 3: “Statistical testing and strength:

Your reported accuracy (95.94%) lacks statistical support as a confidence interval or p-ma. Think of multiple tests and reporting of standard deviations or importance tests to strengthen the reliability of your findings.”

Response 3: Thank you for pointing this out, we sincerely appreciate this suggestion. We added a statistical analysis of data related to standard deviation, confidence intervals, etc. for action recognition. The results confirm the stability and accuracy of our method. For detailed data, please refer to the content of the paper (lines 476-499).

Comment 4: “Missing limitation and future work:

The conclusion was slightly improved, but does not include structured discussion of boundaries or future improvement. Please add a section discussing obstacles (eg sensitivity to noise, fixed camera status) and proposes clear directions for future research.”

Response 4: Thank you for pointing this out, we sincerely appreciate this suggestion. In response to your valuable suggestions, we have completely restructured this section to provide clearer organization and deeper analysis:

Current limitations:

(1) Sensitivity to noise … (2) Fixed Camera Assumptions… (3) The current validation dataset is also limited… (4) Due to some complexity…

Future works:

(1) Improving noise robustness… (2) For distance and range limitations… (3) A broader population… (4) The complexity of the algorithm…

Please refer to lines 547-579 of the paper for details.

We greatly appreciate your suggestions and comments and hope that our revisions and responses will satisfy you.

Thanks again.

Sincerely

Hang Li